# A novel role for Dun1 in the regulation of origin firing upon hyper-acetylation of H3K56

**Lihi Gershon** [ID] **, Martin Kupiec** [ID] *

The Shmunis School of Biomedicine & Cancer Research, Tel Aviv University, Israel

* martin@tauex.tau.ac.il

**Data Availability Statement:** All relevant data are within the manuscript and its Supporting Information files.

**Funding:** MK was supported by grants from the Israel Science Foundation (ISF: https://www.isf.org.

## Abstract

During DNA replication newly synthesized histones are incorporated into the chromatin of the replicating sister chromatids. In the yeast *Saccharomyces cerevisiae* new histone H3 molecules are acetylated at lysine 56. This modification is carefully regulated during the cell cycle, and any disruption of this process is a source of genomic instability. Here we show that the protein kinase Dun1 is necessary in order to maintain viability in the absence of the histone deacetylases Hst3 and Hst4, which remove the acetyl moiety from histone H3. This lethality is not due to the well-characterized role of Dun1 in upregulating dNTPs, but rather because Dun1 is needed in order to counteract the checkpoint kinase Rad53 (human CHK2) that represses the activity of late firing origins. Deletion of *CTF18*, encoding the large subunit of an alternative RFC-like complex (RLC), but not of components of the Elg1 or Rad24 RLCs, is enough to overcome the dependency of cells with hyper-acetylated histones on Dun1. We show that the detrimental function of Ctf18 depends on its interaction with the leading strand polymerase, Polε. Our results thus show that the main problem of cells with hyper-acetylated histones is the regulation of their temporal and replication programs, and uncover novel functions for the Dun1 protein kinase and the Ctf18 clamp loader.

## Author summary

Within the cell's nucleus the DNA is wrapped around proteins called histones. Upon DNA replication, newly synthesized H3 histones are acetylated at lysine 56. This acetylation is significant for the cell because when it is not removed in a timely manner it leads to genomic instability. We have investigated the source of this instability and discovered that the kinase Dun1, usually implicated in the regulation of dNTPs, the building blocks of DNA, has a novel, dNTP-independent, essential role when histones are hyper-acetylated. The essential role of Dun1 is in the regulation of the temporal program of DNA replication. Thus, our results uncover what the main defect is in cells unable to regulate the acetylation of histones, while revealing new functions for well-characterized proteins with roles in genome stability maintenance.

il), the Israel Cancer Research Fund (ICRF: https://www.icrfonline.org) and the Minerva Stiftung (https://www.minerva.mpg.de). The funders had no role in study design, data collection and analysis, decision to publish, or preparation of the manuscript.

**Competing interests:** The authors have declared that no competing interests exist.

## Introduction

Control over the chromatin state in living cells is essential for viability and continuous growth. The maintenance of epigenetic memory requires that upon DNA replication the information present as histone modifications be transmitted to the newly synthesized chromatid, so the two cells emerging from the process possess identical modifications. As cells replicate their DNA, old nucleosomes located before the fork are transferred to the newly synthesized DNA behind the fork. As the amount of DNA has doubled, newly synthesized histones are incorporated together with the old ones. In order to facilitate this process, the newly synthesized H3 histones are flagged: they undergo a chain of handoffs starting with the histone chaperone Asf1, who delivers the histones to the acetyltransferase Rtt109 for acetylation at lysine 56 of histone H3 (H3K56) [1]. Following this, the dimer composed of H3K56Ac and histone H4 is handed off to the chromatin remodelers CAF-1, Rtt106 and the FACT complex, in order to be incorporated into the assembling chromatin [2].

H3K56 acetylation is cell cycle dependent, and in unperturbed cell cycles acetylated H3K56 accumulates during S phase and is removed at $G_2$/M by the histone deacetylase Hst3 and at M/$G_1$ by its paralog Hst4. If the cell cycle is perturbed (for example, by DNA damage) K56 may remain acetylated for longer time [2,3]. Hst3 plays a more prominent role than Hst4 in the deacetylation of K56 [4]. In double mutant *Δhst3 Δhst4* cells, H3K56 is acetylated on all histones and throughout the entire cell cycle. This leads to a plethora of genomic instability phenotypes and to temperature sensitivity (TS) [5]. On the other hand, the process of acetylation seems to be needed, as cells unable to acetylate histone H3 on lysine 56 (e.g. due to deletion of Asf1 or Rtt109, or to expression of the non-acetylatable allele H3K56R as the only source of histone H3) also exhibit genome instability phenotypes [6].

The Dun1 protein kinase is part of the S-phase checkpoint, which is activated when replication forks are stalled. Upon encountering DNA damage during the S-phase, the sensor Mec1 (Mammalian ATR) is recruited to ssDNA regions and phosphorylates the effector Rad53 (Chk2) with the help of two mediator proteins, Mrc1 (Claspin) and Rad9 (53BP1) [7]. This results, among other processes, in stabilization of the replication forks, repression of late firing origins and an elevation of dNTP pools in the cell [8–10]. Dun1 plays a double role in the upregulation of dNTPs synthesis by the ribonucleotide reductase (RNR) complex [11]. It inhibits the repressor Crt1/Rfx1, thus elevating the transcription of the *RNR2*, *RNR3* and *RNR4* genes, encoding subunits of RNR [12]. In addition, Dun1 phosphorylates the Rnr1 inhibitor Sml1, sending it to degradation [13] thus increasing RNR activity. Although there is ample evidence for additional functions of Dun1, information pertaining to Dun1p's roles outside of nucleotide pools regulation is scarce [14].

Mrc1 is a protein with two functions during DNA replication. In addition to its central role in the DNA replication checkpoint [15], Mrc1 can also act within the replisome itself, where it is found in a complex with Tof1 and Csm3 [16]. Mrc1 touches the replisome at several places: it interacts with Tof1, with the replicative helicase, and, indirectly, with Pol2, the catalytic subunit of Polε [17]. Upon DNA damage, Mrc1 couples with Tof1 to form a stable pausing complex that protects paused forks from collapse [18]. Finally, it was found that Mrc1 is needed in order to maintain fork progression rate, and in its absence forks advance more slowly [19].

Rfc1 is an essential protein that, together with the Rfc2-5 subunits, forms a pentameric complex that loads the PCNA ring onto the chromatin during DNA replication. Three additional RFC-like complexes (RLCs) have been characterized, in which the same Rfc2-5 subunits interact with either Elg1, Ctf18 or Rad24 [20]. RFC$^{CTF18}$ is unique amongst the RLCs as instead of creating a pentameric ring, it forms a heptameric ring with two additional proteins, Dcc1 and Ctf8. The RFC$^{CTF18}$ RLC was shown to have ATPase-dependent PCNA loading and

unloading abilities *in vitro* [21]. Recently, it has also been shown to load PCNA in order to promote sister chromatid cohesion (SCC) [22], and indeed in the absence of Ctf18 there is a prominent cohesion defect [23]. Ctf18 RLC mutants are defective for the intra-S checkpoint, exhibiting abundant firing of late origins upon depletion of dNTPs by treatment with hydroxyurea (HU). This result was shown to be independent of Ctf18's role in SCC maintenance [24].

Cells begin replication from conserved sites along the genome called origins of replication. Following the assembly of the pre-RC (licensing) complex, there is sequential activation of two central protein kinases: DDK (Dbf4 Dependent Kinase) which phosphorylates the MCM2-7 helicase, allowing the loading of the replisome proteins Cdc45 and Sld3; and CDK (Cyclin Dependent Kinase), which enables the recruitment of the GINS complex, Dpb11, Sld2 and Polε (pre-LC). Their concerted action allows DNA replication firing. In order to guarantee that the genome will be replicated only once, the proteins Dbf4, Dpb11, Sld2 and Sld3 are found at low levels in the cell, and their activation by the CDK and the DDK is temporally separated from the activation of the MCM hexamer [25,26]. Sld3 and Dbf4 are also targets of Rad53, whose role in the checkpoint is to stabilize stalled replication forks and inhibit further firing from late-replicating origins [27].

Here we bring evidence that upon hyper-acetylation of H3K56, the kinase Dun1 carries out an essential role that is unrelated to its role in the maintenance of dNTP levels. We show that Dun1 performs its roles independently of histone deposition and that the synthetic lethality stemming from the lack of Dun1 can be alleviated by deletion of the alternative clamp loader *CTF18* but not by deletion of the other alternative clamp loaders, *ELG1* and *RAD24*. Our work unveils that the detrimental activity of Ctf18 is independent of its roles in PCNA loading/unloading or in SCC and depends on Ctf18's association with the Pol2 subunit of Polε. We show that abolishing Rad53's repression of late origins is enough to rescue the lethality of *Δdun1*, indicating that Dun1 is needed in order to counteract Rad53's action on firing factors of late-firing origins.

## Results

### The lethality of *Δdun1* in the presence of hyper-acetylation of H3K56 can be prevented by deleting *CTF18*

In order to analyze the cellular roles played by histone acetylation, we grew cells deleted for the *HST3* and *HST4* genes and carrying a *URA3*-marked centromeric plasmid bearing the *HST3* gene, and tested the ability of different strains to grow on plates containing 5-FOA, which select for cells that have lost the plasmid and therefore suffer from hyper-acetylation of H3K56. Serial dilution plating of these cells showed that the *Δhst3 Δhst4* strains are able to grow at 25˚C, but are unable to thrive at temperatures above 30˚C (Fig 1A). We confirmed previous results [4] showing that mutations in the yeast RFC-like complexes (RLCs: *elg1*, *ctf18* or *rad24*) suppress the inability of the *Δhst3 Δhst4* mutants to grow at high temperatures (Fig 1A). In contrast, the protein kinase Dun1, which is mainly implicated in the regulation of dNTPs, is essential for viability in histone hyper-acetylation conditions even at low temperature (Fig 1A; [4,28]). Similar results were obtained using an unbiased method checking for spontaneous plasmid loss. Strains lacking the *ADE2* gene are red, due to the accumulation of a red pigment in the cells. Mutations in *ADE3*, which acts upstream of *ADE2*, prevent pigment accumulation resulting in white colonies. Thus, the presence of *ADE3* on the plasmid containing the *HST3* gene complements the *Δade3* mutation and confers a red color, which is lost if the cells lose the *HST3*-containing plasmid [29]. Whereas 12% of the *Δhst3 Δhst4* cells were able to lose the plasmid, less than 1% of *Δhst3 Δsht4 Δdun1* cells were able to do so. We used this system as a means to verify that our results are not the product of mutations in the *URA3*

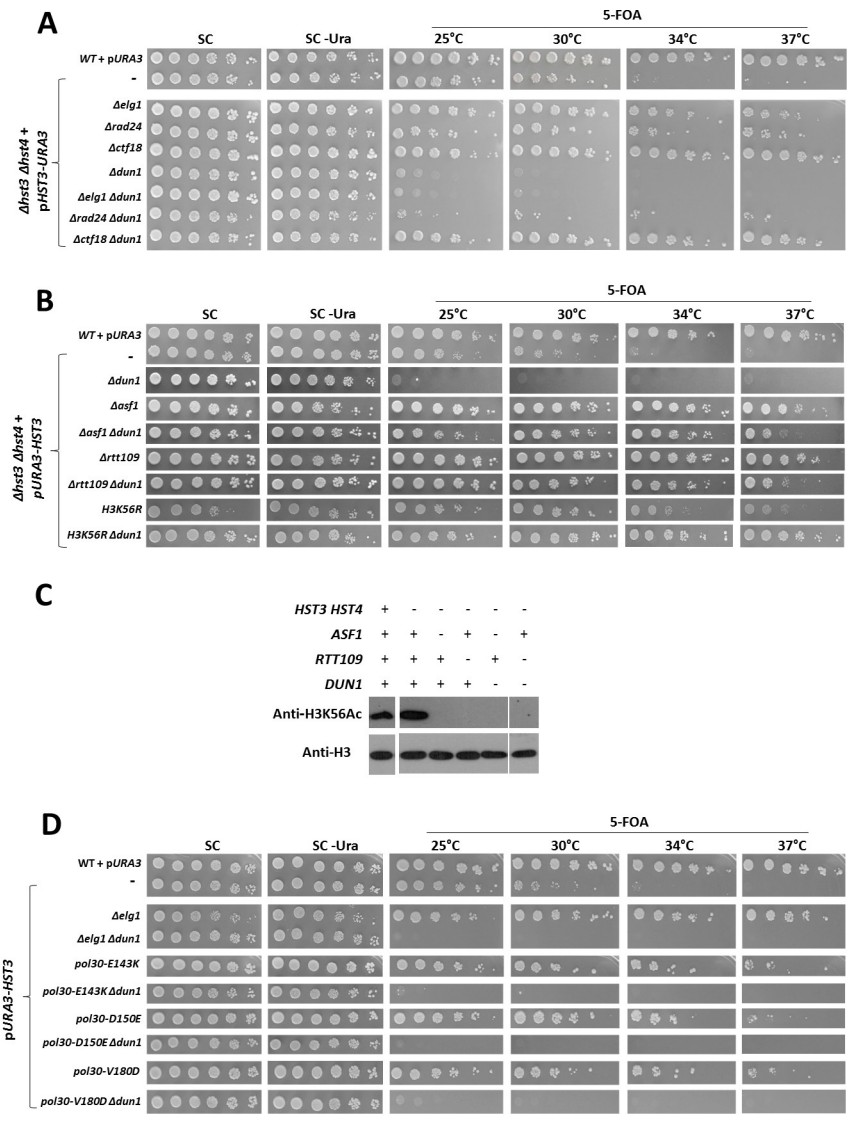

**Fig 1. Dun1 is necessary for the viability of *Δhst3 Δhst4* cells, while Ctf18 is detrimental. (A)** Serial dilutions of various strains were plated on Synthetic Complete (SC) medium, SC lacking uracil at 30°C, or on plates containing 5-FOA at the indicated temperatures, for 3 days. The wild type strain carried an empty *URA3*-marked plasmid (pRS315). All the other strains were *Δhst3 Δhst4* and carried a pRS315 plasmid with the *HST3* gene (p*URA3-HST3*). Additional mutations in each strain are indicated. **(B)** The lethality of *Δdun1* is abolished by mutations that prevent H3K56 acetylation. Mutating lysine 56 to arginine also rescues the lethality of *Δdun1*, when cells are deleted for *HHT1-HHF1* and *HHT2-HHF2* and express the H3K56R as their only H3 copy from a centromeric plasmid **(C)** Deletion of the histone chaperon Asf1 and the acetyl transferase Rtt109 abolish H3K56 acetylation. Mid-log cells were harvested, protein extracted and proteins were imaged using an antibody against H3K56Ac or against unmodified histone H3. **(D)** Reducing or increasing PCNA levels on chromatin does not mimic lack of Ctf18. See text for explanation.

gene on the *HST3*-containing plasmid or due to the use of 5-FOA (S1A Fig). The synthetic lethality was further verified using tetrad analysis (S1B Fig).

The dependency on Dun1 for viability is directly related to the acetylation state of the cell, as no lethality is observed if acetylation at K56 of histone H3 is prevented by deleting the gene encoding the acetyl transferase Rtt109 or the histone chaperon Asf1 or by mutating the K56 residue to arginine (Fig 1B and 1C). Strikingly, the need for Dun1 activity can be suppressed

by deletion of *CTF18* (Fig 1A), but not by deletion of any of the two remaining RLCs (Fig 1A). This suggests a unique role for Ctf18 in cells with hyper-acetylated H3K56.

We sought to determine if Ctf18's detrimental function is related to its potential PCNA loading or unloading activity. We reasoned that if Ctf18's role is related to the amount of PCNA on the chromatin, then reducing or increasing the level of PCNA on the chromatin would mimic its deletion. Increasing the amount of PCNA on chromatin by deletion of the PCNA unloader *ELG1* [30–32] (Fig 1C) or by PCNA overexpression from a high-copy-number plasmid does not suppress the lethality of *Δdun1 Δhst3 Δhst4* cells. Similarly, reducing the amount of PCNA in *Δdun1* strains by introducing PCNA mutants that spontaneously dissociate from chromatin, such as *pol30-E143K*, *pol30-D150E* or *pol30-V180D* [33] did not result in suppression (Fig 1D). These results suggest that changes in the level of PCNA are not the root cause for the lethality of *Δdun1* cells with hyper-acetylated histones, nor the mechanism by which deletion of *CTF18* suppresses it. Collectively, our findings point to a fundamental role for Dun1, which promotes the viability of cells exposed to hyper-acetylation of H3K56, and depends on an activity of Ctf18 that is not mediated by the levels of PCNA on the chromatin.

## A dNTPs-independent role for Dun1 in maintaining viability of cells experiencing hyper-acetylation of H3K56

Dun1 participates in the S-phase checkpoint signaling cascade [13,16]. Dun1's most prominent role is the upregulation of dNTPs by removing the inhibition on the expression and activity of subunits of the RNR complex mediated by Sml1, Dif1 and Crt1 (Fig 2A). This is done

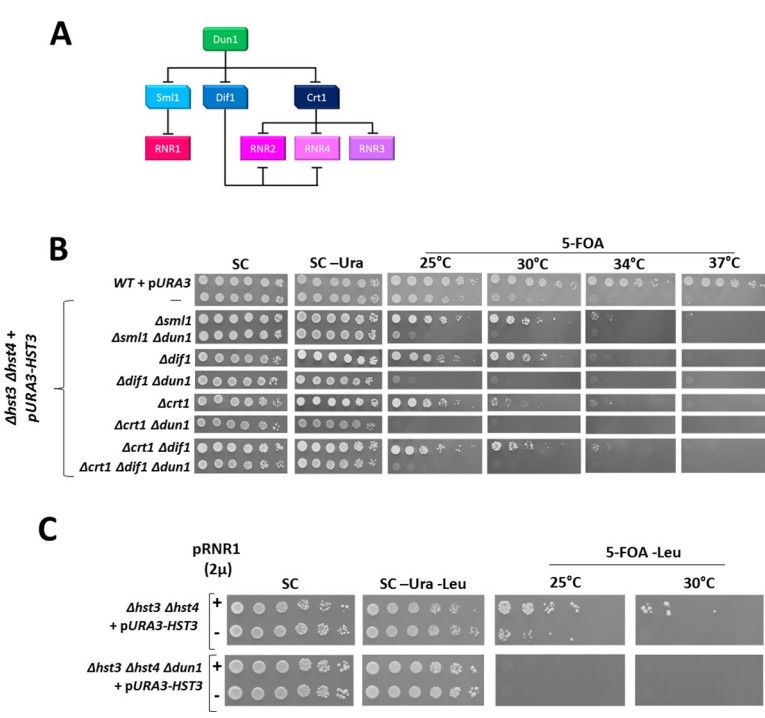

**Fig 2. Dun1's vital function upon hyper-acetylation of H3K56 is independent from its role in regulating the dNTP levels of the cell. (A)** A schematic representation of the way Dun1 regulates dNTPs. Dun1 negatively controls *SML1*, an inhibitor of *RNR1*, and the repressors of *RNR2*, *RNR3* and *RNR4*, Crt1/Rfx1 and Dif1. **(B)** Increasing the levels of dNTPs by deleting the inhibitors of the RNR complex does not rescue the lethality of *Δdun1* cells. Serial dilutions were plated as in Fig 1A. **(C)** Increasing dNTP levels by over-expressing Rnr1 does not rescue the lethality of *Δdun1* cells. Cells were transformed with a high copy number plasmid carrying, or not, *RNR1*, the large subunit of the RNR complex. Serial dilutions were plated as in Fig 1A.

following a signal transduced to Dun1 by Rad53 in response to the activation of the checkpoint due to DNA-related stress [34]. Previous work has established that upon entrance into S-phase the level of dNTPs increases ~3-fold [35]. Deletion of the RNR inhibitor *SML1* leads to a similar increase in dNTP levels [36]. We reasoned that since most of Dun1's characterized roles are linked, directly or indirectly, to dNTPs, bypassing the regulation that Dun1 exerts on the RNRs could restore viability to *Δdun1* strains. However, the lethality of *Δdun1* cells was not suppressed by deleting the RNRs' negative regulators: *CRT1*, *DIF1* and *SML1*, individually or together (Fig 2B), showing that Dun1's essential role is unrelated to dNTP levels in the cell. Moreover, overexpression of RNR1 was also unable to rescue the *Δdun1* lethality (Fig 2C), despite elevating the dNTP levels by 10 fold [37]. These findings point to a novel role for Dun1 that is independent of its well-established role in the regulation of dNTP levels. This function becomes essential for viability when cells experience hyper-acetylation of H3K56.

## Dun1 is not directly involved in histone deposition

We turned our attention to the histone deposition pathway. One way in which Dun1 may work to promote viability in the presence of hyper-acetylation of H3K56 would be to alter the amount of acetylated histones deposited onto the chromatin. Within the nucleus there are two sources of H3 histones: cells can transfer parental histones present in front of the fork to the newly replicated DNA that is being created behind it. This mechanism is dependent on the Dpb3, Dpb4 and Mcm2 proteins [38,39]. However, as the amount of DNA behind the fork is twice that at its front, additional histones are needed. The insertion of the H3K56 acetylated, newly synthesized histones relies on the CAF-1 and FACT complexes as well as on Rtt106 [40–42] (Fig 3A). In cells lacking *HST3* and *HST4*, H3K56Ac histones will be found both in front of, and behind, the replication fork. Since we cannot directly work on strains that are both hyper-acetylated at H3K56 and lack *DUN1*, we created *Δhst3 Δhst4* strains with mutations in various chromatin-affecting proteins and followed their phenotype. If Dun1 interacts with these proteins in a positive way and is involved in the re-positioning of the histones, then introducing mutations into these proteins should be lethal as well. Conversely, if Dun1 negatively regulates the re-positioning process that becomes detrimental upon hyper-acetylation of H3K56, then deleting the targets of regulation should restore the viability to *Δdun1* cells (as we have observed for *Δctf18*, for example). In the case of FACT we used an allele of Spt16 defective in the binding to newly synthesized H3 histones [42] and in the case of *MCM2*, which encodes another essential protein, we used an allele defective for the transfer of histones from the front to behind the fork [43]. We were most interested in the effect of *Δrtt106* and *Δcac1* as there is evidence to suggest that they are important for the stabilization of replication forks [44]. However, our results clearly show that alone or in various combinations, none of these chromatin remodelers is essential for viability in the presence of hyper-acetylation of H3K56, nor does their deletion restore viability to *Δdun1* (Fig 3B and 3C). Taken together, these results argue that the lethality of *Δdun1* is not due to defects in components connected to chromatin remodeling or histone processing.

## Involvement of checkpoint proteins in maintaining viability in the presence of hyper-acetylation of H3K56

Previous work has shown that cells that experience hyper-acetylation of H3K56 also exhibit spontaneous activation of Rad53 and are dependent on the checkpoint kinase Mec1 (ATR) for survival [4], suggesting a model in which checkpoint activation becomes essential for life. According to this model, *Δhst3 Δhst4* cells would die in the absence of any of the signal transduction components: Mec1, Rad53, Rad9 or Mrc1. For this reason, we decided to look at the

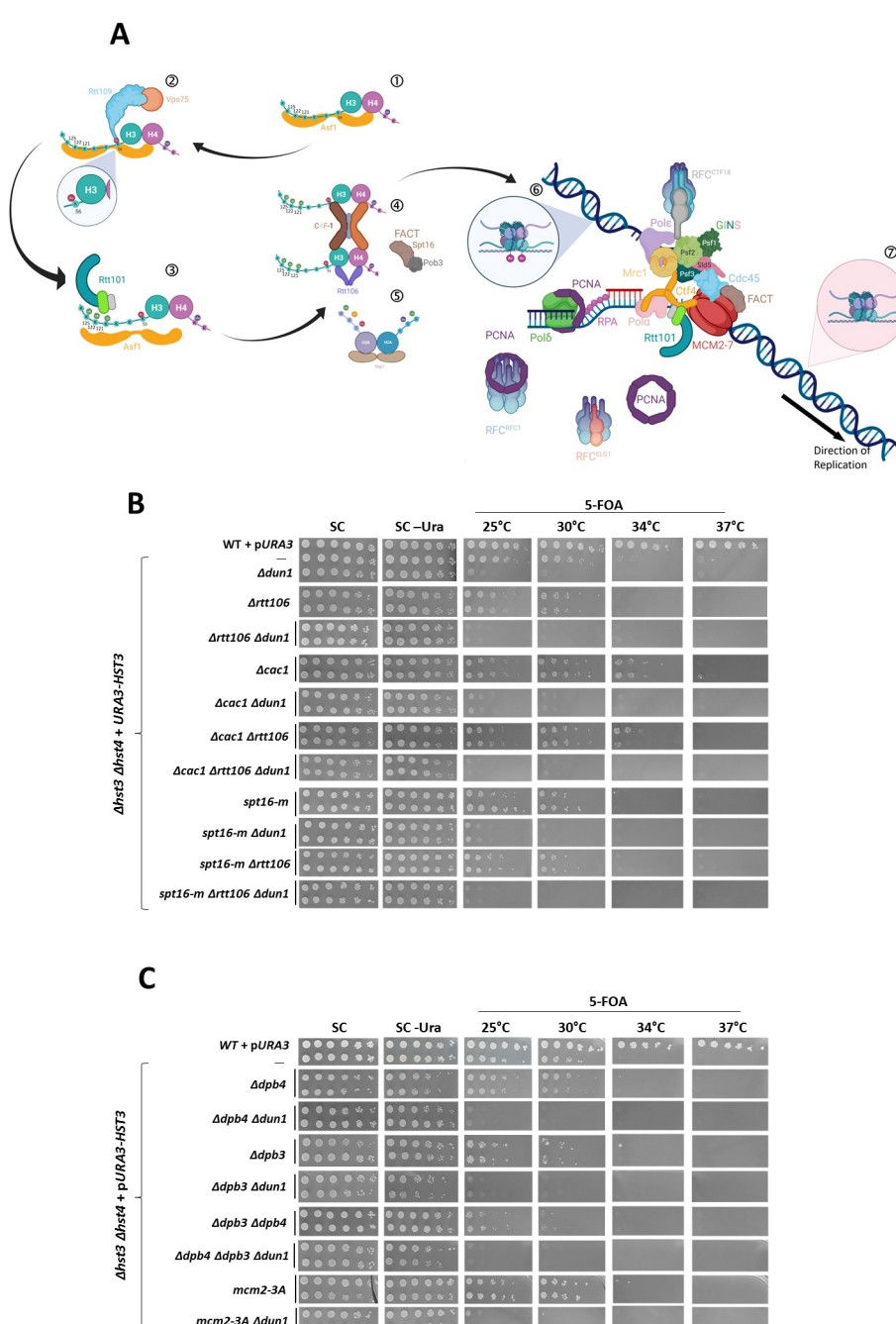

**Fig 3. Dun1 is not directly involved in histone deposition. (A)** A schematic representation of the path of newly synthesized H3 histones: 1) Asf1 binds newly synthesized H3-H4 histones. 2) Asf1 presents the H3-H4 dimer to Rtt109 to be acetylated at H3K56. 3) K56 acetylation increases the affinity of Rtt101 to the histones, causing it to ubiquitinate the H3K56Ac histone. 4) Ubiquitination facilitates the detachment of H3K56Ac-H4 from Asf1 and its handoff to the chromatin remodelers. 5) H2A and H2B are brought to the chromatin by Nap1. 6) H3K56Ac histones are incorporated behind the replication fork. New histones mixed with parental histones are transferred back by the replication machinery. 7) In front of the replication fork all histones lack acetyl groups due to the activity of Hst3 and Hst4 at the previous cell cycle. **(B, C)** Dun1 is not involved in histone deposition by the replication machinery. Mutants affecting histone deposition do not mimic or affect the phenotype of *Δdun1* cells. Serial dilutions were plated as in Fig 1A.

components of the checkpoint and their behavior upon exposure to DNA damage. The results, however, were complex: A *Δmec1 Δsml1* strain is lethal in the presence of hyper-acetylated histones, and this lethality, similar to that of *Δdun1*, can be rescued by deletion of *CTF18* (Fig 4A). In contrast, deletion of the Rad9 adaptor (mammalian 53BP1) is not lethal, and can suppress the TS phenotype of *Δhst3 Δhst4* (Fig 4B [45]), indicating that not all the checkpoint components are essential. Moreover, a *RAD9* deletion was unable to suppress the lethality of *Δdun1* (Fig 4B). Deletion of the second adaptor, Mrc1, also resulted in lethality (Fig 4B). However, Mrc1 has roles not only in checkpoint activation. It can also be found at the replisome [17]. Two alleles of Mrc1, termed *mrc1-AQ* and *mrc1-C14*, separate between these functions. The *mrc1-AQ* allele has all 17 SQ/TQ sites (recognized by Mec1) mutated to alanine, effectively abolishing Mrc1's signaling function. The *mrc1-C14* allele is a truncation of the protein, which ends at amino acid 843, and is able to activate Rad53 but exhibits a delayed and expanded S phase similarly to a full deletion of *MRC1*, indicating that this allele is defective in a replicative function [46]. We found that while the *mrc1-AQ* allele has no effect on the ability of *Δhst3 Δhst4* to grow, the *mrc1-C14* replicative allele conferred, upon hyper-acetylation of H3K56, the same lethality as the deletion of the entire gene (Fig 4C). These results indicate that the essential function of Mrc1 in the presence of hyper-acetylated histones is related to a function carried out at the replisome. Finally, when we deleted Rad53, we discovered that it was able to suppress the lethality of *Δdun1*, although not that of *Δmec1* (Fig 4A). The double mutant *Δmrc1 Δrad9* is inviable in strains with hyper-acetylated histones (Fig 4B), despite the fact that in it no activation of Rad53 takes place [47], and thus would in principle be expected to show a phenotype similar to that of *Δrad53*. This is probably due to replicative functions carried out by Mrc1 that prevent viability to such a strain. Consistent with this idea, deletion of *RAD53* in the background of *Δmrc1* did not restore viability to *Δmrc1* cells.

Since Rad53 activity appeared to be detrimental for *Δhst3 Δhst4* strains, we asked whether Ctf18 is capable of rescuing the lethality of *Δdun1* by preventing activation of Rad53. We checked the activation of the checkpoint in strains carrying various mutations, alone or in combinations (Fig 4D). We observed only minor differences in the ability of the different mutants to phosphorylate Rad53 upon MMS treatment. This led us to conclude that Rad53 activation is not prevented in strains lacking Ctf18, and that Rad53 phosphorylation was not the cause of the phenotypes observed in the absence of Dun1.

## The essential function of Dun1 in cells carrying hyper-acetylated histone 3 requires its phosphorylation by Rad53

The results presented so far led us to speculate that Mec1 might activate Dun1 independently of Rad53. It is conceivable that under conditions of hyper-acetylation of H3K56, Rad53 sends out a detrimental signal to an unknown target, whereas Mrc1 and Dun1 play essential roles that maintain viability, and are independent of Rad53. This hypothesis is supported by the fact that a strain lacking both *RAD9* and *MRC1* is still dead in the presence of hyper-acetylated histones, despite lacking Rad53 activation (Fig 4B) [16]. If our hypothesis was correct, then there should be activation of Dun1 even in the absence of Rad53. In order to examine this possibility, we aimed at abolishing the interaction between the two proteins. Rad53 and Dun1 both contain Forkhead Associated (FHA) domains, while Rad53 alone contains also two SQ/TQ clusters (Fig 5A). Rad53 binds to Dun1 using Rad53's first SQ/TQ Cluster Domain (SCD1), and in it the most important residues are T5 and T8. Two arginines, R60 and R62 of Dun1, bind T5 of Rad53; K100 and R102 of Dun1 bind T8 of Rad53; S74 and H77 are conserved residues of FHA domains across *S. cerevisiae* and K129 stabilizes the binding of Dun1 to Rad53's first SCD. Finally, residues S10 and S139 are autophosphorylation sites (Fig 5A) [48]. We

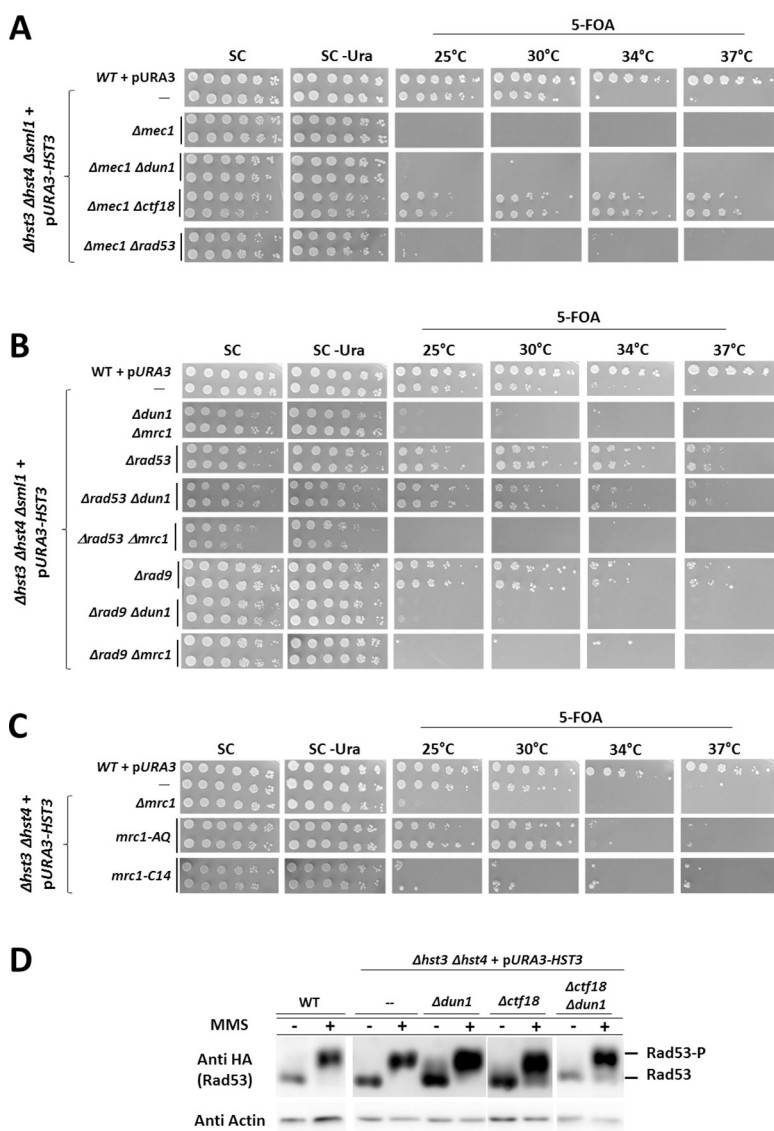

**Fig 4. S-phase checkpoint proteins affect cells that are hyper-acetylated at histone H3K56 in different ways. (A)** Mec1 is necessary for viability, but can be rescued by deletion of *CTF18*. **(B)** The lack of Mrc1 leads to lethality. Deletion of *RAD53* suppresses the lethality of *Δdun1*, but not of *Δmrc1*. Deletion of *RAD9* suppresses the temperature sensitivity of *Δhst3 Δhst4* mutants but is not lethal nor does it rescue the lethality of *Δdun1* mutants. A combined deletion of both *MRC1* and *RAD9* (in the background of *Δsml1*), which is defective in Rad53 activation, is inviable upon hyper-acetylation. **(C)** Mrc1's checkpoint function is dispensable, while its replicative function is essential. **(D)** Deletion of *CTF18* does not prevent checkpoint activation upon 0.01% MMS treatment for 1 hour. Serial dilutions were plated as in Fig 1A.

mutated all these sites, or subsets of them. The combinations shown in Fig 5B were examined using Phos-tag protein gels. To prevent lethality, all strains carried the covering *URA3-HST3* plasmid. As can be seen, abolishing all interaction sites together (*dun1-9A*) completely abolishes Dun1's ability to react to DNA damage created by MMS, as does deletion of Rad53. This demonstrates a complete dependence of Dun1 on Rad53 for activation. Abolishing only some of the sites (*dun1-4A*) reduces the phosphorylation upon MMS treatment, but does not abolish it, while abolishing Dun1's autophosphorylation sites (*dun1-APM*) has a very mild effect. Mutations in Mrc1 also presented a pattern of Dun1 phosphorylation similar to that of the

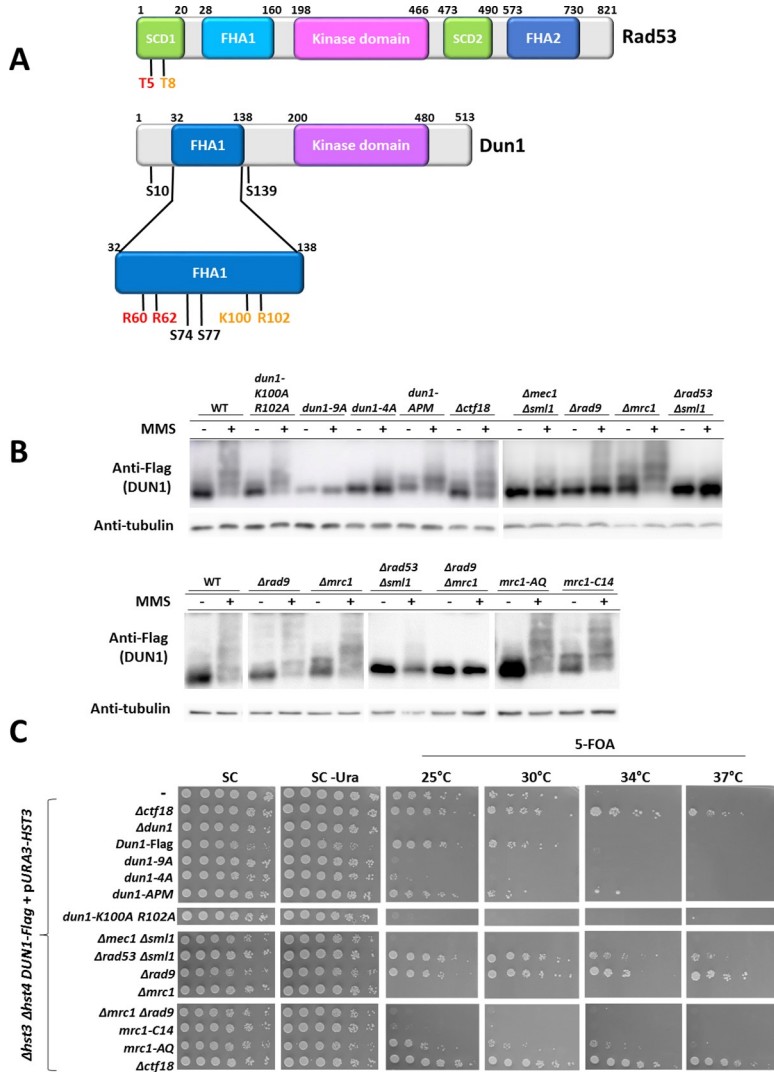

**Fig 5. Dun1 must be phosphorylated by Rad53 in order to perform its viability promoting function. (A)** Schematic description of Rad53 and Dun1's domains and relevant amino acids. Rad53 and Dun1 both contain Forkhead Associated (FHA) domains, while Rad53 alone contains also two SQ/TQ clusters. Rad53 binds to Dun1 using Rad53's first SQ/TQ Cluster Domain (SCD1), and in it the most important residues are T5 and T8. Two arginines, R60 and R62 of Dun1, bind T5 of Rad53; K100 and R102 of Dun1 bind the T8 residue of Rad53. In addition, S74 and H77 are conserved residues of FHA domains across *S. cerevisiae* and K129 stabilizes the binding of Dun1 to Rad53's first SCD. Finally, serines S10 and S139 are autophosphorylation sites. **(B)** Mid-log cultures were exposed, or not, to MMS (0.01%) for an hour before proteins were extracted and loaded onto Phos-Tag-containing SDS-PAGE gels. Proteins were imaged using an antibody against the Flag tag of the *DUN1* alleles in the strains used. **(C)** The phosphorylation state of Dun1 is not indicative of viability. The same strains as in **(B)** were subjected to a serial dilution assay.

WT. The small amount of phosphorylation observed in the *Δmec1* strain is consistent with Rad53 having a role in the cell cycle that leads to it being phosphorylated in a Mec1-independent manner [49].

Next, we looked at the viability of these Dun1 mutants upon hyper-acetylation of H3K56 (upon loss of the *HST3*-bearing plasmid). If some sites are used to promote viability while others, when phosphorylated, promote detrimental functions, then only some of the mutants should be viable. Surprisingly, almost all mutants in the Rad53 contact sites of Dun1 were lethal, despite the fact that lack of Rad53 rescues the lethality of a *Δdun1* strain (Fig 5C). We

observed Dun1 mutants that are phosphorylated and inviable (e.g.: *dun1 K100A R102A* and *dun1-4A*), while a *Δrad53* mutant has no phosphorylation and yet it is alive. We thus conclude that Dun1's phosphorylation does not affect its ability to promote viability under hyper-acetylated histone conditions. While we cannot exclude the possibility that other factors might interact with Dun1 using these residues, we favor a model in which Dun1's activity is needed only as long as Rad53 is also active. In such a case, Rad53 is a double edged sword–it is needed in order to activate Dun1 and preserve viability but at the same time activation of a different target by Rad53 in a *Δhst3 Δhst4* strain has detrimental effects that are counteracted or prevented by Dun1.

## Ctf18's association with Polε is detrimental in the presence of hyper-acetylated histones

Recently, Ctf18 was also found to be connected to the replisome, and several papers point out that the Ctf18 RLC forms a physical connection with Polε via its Dcc1 subunit [50]. An allele of Ctf18, *ctf18-2A*, in which the conserved tryptophans 736 and 740 are replaced by alanines, is incapable of interacting with Dcc1, effectively abolishing the connection between Ctf18 and Polε [51]. Fig 6A shows that the *ctf18-2A* strain has the same phenotypes as the *Δctf18* strain. Thus it is enough to prevent Ctf18's interaction with the polymerase to completely rescue the lethality of both *Δdun1* and *Δmrc1* in cells with hyper-acetylated histones. The *ctf18-2A* allele is expressed at similar levels as the WT *CTF18* allele (Fig 6B). Thus, Ctf18's detrimental function is carried out via its connection to the leading strand polymerase.

While there is evidence that both Ctf18 [23] and Polε [52] are involved in SCC establishment and maintenance, we were able to exclude the possibility that Dun1 and Mrc1's role upon hyper-acetylation of H3K56 is related to SCC. By both over-expressing Mcd1/Scc1, the alpha-kleisin subunit of the cohesin complex, whose cleavage causes the dissociation of cohesin from the chromosomes at anaphase, and by deleting *WPL1/RAD61*, which is a negative regulator of cohesion [53], we show that increased stability of the cohesin complex is incapable of imitating the *Δctf18* phenotype or to rescue the *Δdun1* phenotype (Fig 6C and 6D).

Taken together, our results show that the synthetic lethality of *Δhst3 Δhst4* with *Δdun1* can be alleviated by eliminating the activity of the Ctf18 RLC, or simply by detaching it from the leading strand of the moving fork. The negative function of Ctf18 is unrelated to its role in maintaining SCC or in depositing PCNA onto the chromatin.

## Dun1's role is in promoting firing of late origins

Upon activation, Rad53 prevents the firing of late origins, leading to an arrest in the cell cycle [27]. By mutating Sld3 at 38 sites and combining this mutant with a Dbf4 allele mutated at 4 sites, it is possible to negate the effect of Rad53 on these proteins, causing them to activate origins regardless of the input of Rad53. This effect is seen as late origin de-repression upon checkpoint activation [54]. When we combined these alleles into our system, we saw that de-repression of late firing origins rescued the temperature sensitivity of *Δhst3 Δhst4* to the same extent as did the deletion of *CTF18*. It also rescued the lethality of the *Δdun1* strain (Fig 7A), and point to a possible role for Dun1 in negating Rad53's effects on origins, which is detrimental in the presence of hyper-acetylation.

In order to examine this possibility we performed qPCR to check the ability of origins to undergo firing. We created a strain carrying a genomic auxin-inducible degron of Dun1 and examined the ability of two locations in the genome to undergo replication in the presence of the ribonucleotide reductase inhibitor hydroxyurea (HU). HU causes a depletion of dNTP pools, which causes a slowing down of replication progression but does not alter the order in

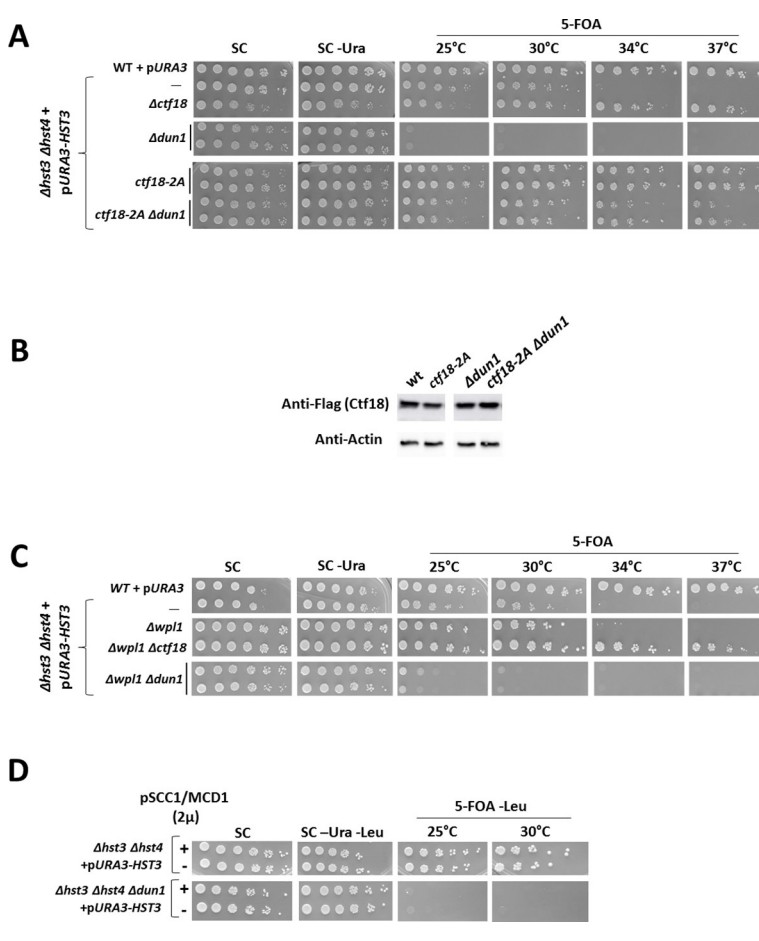

**Fig 6. Ctf18's detrimental activity is mediated by its connection to the Pol2 subunit of the leading strand polymerase Polε while its role in SCC is dispensable. (A)** Mutations in *CTF18* that prevent binding to Polε suppress the lethality of *Δdun1*. **(B)** The *ctf18-2A* allele is expressed at similar levels as the WT *CTF18*. Mid-log cells were harvested, protein extracted and proteins were imaged using an antibody against the Flag tag of the *CTF18* alleles in the strains used. **(C)** Increasing the stability of the cohesin complex by deleting the negative regulator of the cohesin complex *WPL1* does not rescue the lethality of *DUN1* deleted cells. Serial dilutions were plated as in Fig 1A. **(D)** Increasing the stability of the cohesin complex by over expressing the Scc1/Mcd1 subunit of the cohesin complex does not rescue the lethality of *DUN1* deleted cells.

which origins fire [55]. One of the locations examined is the efficient early firing origin ARS305, located on chromosome 3. This origin fires in most cell cycles, and shows activation as early as 10 minutes into the cell cycle in replication progression studies [55]. The other location is the inefficient ARS14-705 located at the proximal right end of chromosome XIV. This origin is known to fire sporadically and late, showing activation only 25 minutes into the cell cycle [55]. In HU-treated cells, this origin is silenced by Rad53, but was found to be derepressed in *Δhst3 Δhst4* cells [56]. We therefore examined by qPCR the ability of cells with, or without, hyper-acetylation of H3K56 to replicate both areas, and also examined whether this ability depends on the presence of Dun1. The amount of PCR amplification with primers specific for ARS305 or ARS14-705 was compared to that of a late-replicating region of chromosome V (NegV) 80 minutes after HU addition. As can be seen (Fig 7B), the efficient early origin ARS305 is activated in a similar manner regardless of the acetylation state of the cell or of the presence of Dun1. On the other hand, ARS14-705, which is derepressed in *Δhst3 Δhst4* [56], does not fire in the absence of Dun1. The firing of this dubious origin is also dependent

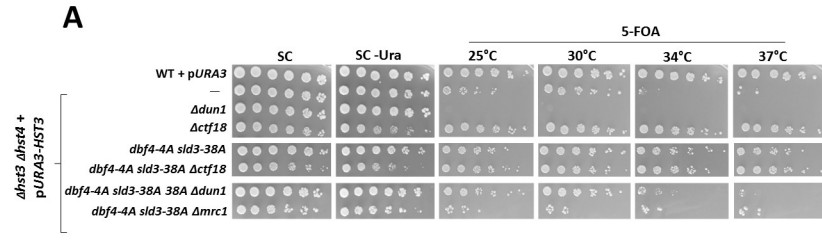

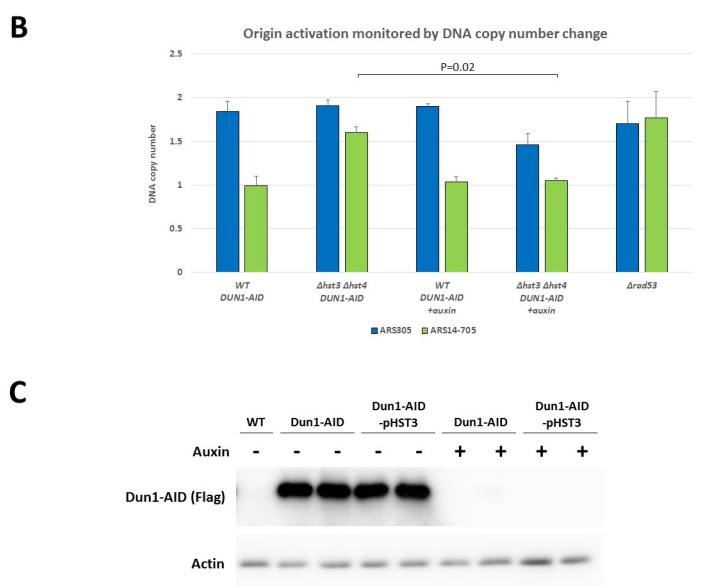

**Fig 7. Dun1's vital role is to negate Rad53's repression of late origins. (A)** Dun1 counteracts Rad53's repression of late firing origins. Preventing Rad53's repression of late firing origins is enough to rescue the lethality of *Δdun1* cells, but not that of *Δmrc1*. Serial dilutions were plated as in Fig 1A. **(B)** Dun1 is needed for late origin ARS14-705 to fire when cells are hyper-acetylated. Dun1 was degraded using auxin. Bars showing the DNA copy number relative to $G_1$ phase 80 minutes following the release of cells into HU. **(C)** Degradation of Dun1 using an auxin inducible degron, monitored prior to the addition of auxin and prior to the start of the experiment, following a 1.5 hours incubation with auxin.

on Rad53 (Fig 7B, [56]). We thus conclude that the detrimental activity of Rad53 upon hyper-acetylation is the phosphorylation of Dbf4 and Sld3, which prevents the firing of late origins. Our results demonstrate that under these circumstances Dun1 is essential, implicating Dun1 for the first time in the regulation of origin firing.

These results, together with the results obtained with the *sld3-38A dbf4-4A* alleles, means that cells that are *Δhst3 Δhst4* need Dun1 in order to either counteract or accommodate Rad53's effects on late origins. Furthermore, these results illustrate that cells that are *Δhst3 Δhst4* benefit from ignoring Rad53's activation, which in these cells is constitutive.

## Discussion

Despite much effort, our understanding of the roles played by the acetylation of histone H3 at lysine 56 in genome duplication and stability is still tenuous at best. Centrally located, H3K56Ac appears to juggle many functions in the cell as both its presence and its absence leads to genomic instability. Here we show that upon perturbation of the de-acetylation

machinery, the protein kinase Dun1 is needed for viability. Our genetic and molecular results indicate that Dun1 carries out an essential function in cells with hyper-acetylated histones, and this function counteracts the repression of late origin firing by Rad53. Mutants with hyper-acetylated histones have been shown to have other phenotypes such as high levels of Rad52 foci and high recombination [57], synthetic lethality with *Δrad52* [4] and defective sister chromatid recombination repair [58]. Thus, lack of the Hstt3 and Hst4 deacetylases creates a pleiotropic vulnerability that makes cells be dependent on functions that are non-essential in wt cells (S1 Table).

Dun1's viability-promoting in cells with hyper-acetylated H3K56 is independent of its well-characterized role in regulating the levels of dNTPs in the cell, as we show that increasing the dNTP levels of a *Δdun1 Δhst3 Δhst4* strain is not enough to suppress its lethality. Only a small number of phenotypes caused by deletion of *DUN1* have been shown up to now to be independent of its role in dNTP production. Our results provide evidence for one such role, related to the firing of late origins of replication.

The synthetic lethality caused by *DUN1* deletion can be suppressed by mutating components of the RFC^CTF18 complex. Moreover, the detrimental activity of the RFC^CTF18 in cells with hyper-acetylated histones depends on its association with the leading strand DNA polymerase. The RFC^CTF18 plays central roles in sister chromatid cohesion and genome maintenance [24,59–63]. Although *in vitro* it is able to both load and unload PCNA, its role *in vivo* is still unclear, particularly since its interaction with Pol2 places it at the leading strand front of the moving fork. The number of PCNA molecules in the leading strand should be much lower than that seen in the Okazaki fragments of the lagging strand. It has been recently proposed that the role of the Ctf18 RLC is to promote sister chromatid cohesion by loading PCNA onto the chromatin [22]. We and others have shown that Ctf18's activity is detrimental when cells experience hyper-acetylation of H3K56 (Fig 1A) [4]. Cells that are *Δhst3 Δhst4* exhibit sister chromatid cohesion defects [64], but our results show that manipulations such as strengthening the cohesion by over-expressing Mcd1/Scc1 or by deleting *WPL1* have no effect (Fig 6B and 6D). Similarly, increasing or decreasing the level of PCNA did not affect the Ctf18-mediated lethality of *Δdun1* (Fig 1D). These results indicate that Ctf18's detrimental activity is unrelated to PCNA loading or SCC. Instead, we show that it is mediated by the binding of the RFC^CTF18 to Polε (Fig 6A). Ctf18 was found to be in contact with Polε throughout the cell cycle [65], via the Dcc1 subunit of the RFC^CTF18 complex. Mutating the conserved W736 and W740 residues [62]. of Ctf18 releases it from the leading strand polymerase [62] and is sufficient to completely eliminate the RFC^CTF18 detrimental effect and restore viability to *Δdun1* mutants (Fig 6A).

Rad53 is spontaneously activated in *Δhst3 Δhst4* cells [4]. Analysis of spontaneous suppressors of the temperature sensitive phenotype of *Δhst3 Δhst4* cells revealed that most of them led to reduced Rad53 activation [45]. Indeed, activated Rad53 exerts a negative effect on cells with hyper-acetylated histones, and deletion of *RAD53* results in complete rescue of the synthetic lethality between *Δdun1* and *Δhst3 Δhst4* (Fig 4B). However, abolishing the interactions of Dun1 with Rad53 by mutating the FHA domain of Dun1, rather than mimicking a *RAD53* deletion, is lethal. The simplest explanation for this result is that in addition to the activation of Dun1, which promotes viability, Rad53 has additional target(s) whose action is detrimental for these cells.

We have managed to identify the targets of Rad53 that are needed for viability. Once activated, Rad53 is responsible for preventing late origin firing. It does so by phosphorylating the firing factors Sld3 and Dbf4. This phosphorylation, and thus the inhibition of late origin firing, is independent of Dun1 [54]. In strains with alleles of Sld3 and Dbf4 that cannot be phosphorylated by Rad53, Dun1 was no longer necessary for maintaining viability (Fig 7A), indicating

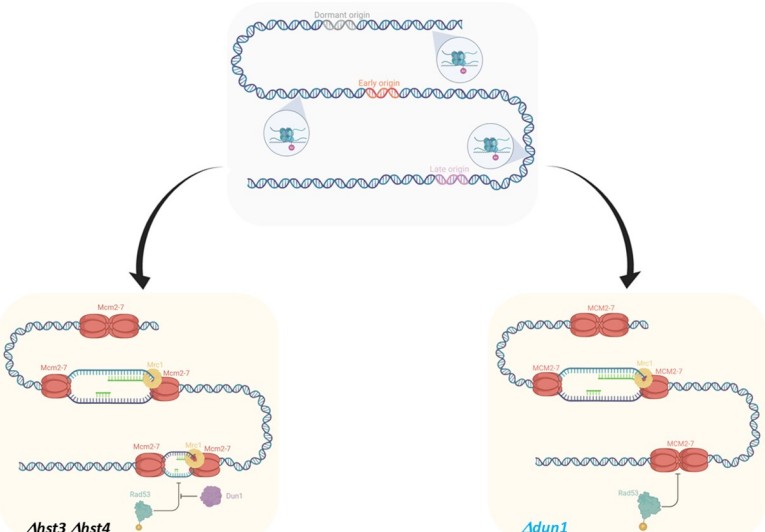

**Fig 8. Harmful effects of hyper-acetylation on replication.** When replication occurs in the presence of hyper-acetylation of H3K56, the replisome must process a histone configuration that is usually only found behind the fork, rather than in front. This creates replication stress, causing Rad53 to be constitutively activated, leading to the suppression of late origins. Dun1 must be activated by Rad53 in order to negate Rad53's activity on late origins, and in its absence forks are forced to travel longer distances, and are more prone to collapse.

that the essential role of Dun1 in *Δhst3 Δhst4* cells is to prevent Rad53's inhibition of late firing origins. Our qPCR analysis of Rad53-dependent late origins known to be de-repressed in *Δhst3 Δhst4* confirms this hypothesis (Fig 7B). Due to the lack of Hst3 and Hst4, there are now H3K56 acetylated histones both in front of, and behind, the replication fork. This is a configuration that replication forks do not encounter in normal S-phase, leading to chronic replication stress, which in turn activates the checkpoint. In this situation, Rad53 is constantly activated and inhibits the firing of most of the late origins, forcing the cells to replicate their genome using forks from early origins (Fig 8). These forks must traverse longer distances, making them sensitive to subtle perturbations in DNA replication and thus more prone to collapse [66,67]. Indeed, it was shown that de-acetylation of H3K56 is important for maintenance of chromosomes carrying long regions without origins [67]. In light of all of our results we propose that Dun1 is responsible for negating the effects of Rad53 on origin firing. There is evidence to suggest that there is some de-repression on late firing origins in *Δhst3 Δhst4* mutants [56], but this de-repression is moderated compared to that observed in a *Δrad53* mutant. By negating some of Rad53's effect on late origins, Dun1 allows firing from a larger number of origins in the later stages of S-phase. Interestingly, although deletion of *MRC1* is also lethal in cells with hyper-acetylated histones, this lethality cannot be suppressed by mutating the phosphorylation sites of Dbf4 and Sld3 (Fig 7B). These results are consistent with the lack of suppression of the synthetic lethality of *Δmrc1* by *Δrad53* (Fig 4B), indicating a different mechanism of action. We have shown that the replicative role of Mrc1, and not its function in checkpoint induction, is essential in cells with hyper-acetylated histones. The high rate of fork collapse that already exists in *Δhst3 Δhst4* cells would become aggravated beyond possibility of repair in the absence of Mrc1, which leads to easy uncoupling of leading and lagging helicases, thus impairing repair and replication re-start.

One of the known phenotypes of *Δctf18* is the de-repression of late origins [24], although the direct mechanism by which Ctf18 ensures late origin repression is unclear. This phenotype is also maintained in cells carrying the allele ctf18-2A, hinting that the de-repression of late

origins might be connected to Ctf18's interaction with Pol2. Given our findings, de-repression of late origins in the absence of Ctf18's interaction with the leading strand polymerase is a possible model to explain why deletion of Ctf18 can rescue the lethality of *Δdun1* in cells with hyper-acetylation of H3 histones. However, it is possible that the connection between Ctf18 and Pol2 also limits the length of replication tracts, possibly in order to periodically deposit a PCNA complex onto the leading strand, a process that is important to keep the sister chromatids together [22]. Indeed, upon Ctf18's deletion there is a marked elongation of replication tracts [68] which can benefit *Δdun1* mutants as it helps in the completion of replication. An alternative third model would suggest a role of the Ctf18 RLC in the transfer of histones from the front to the back of the moving fork. In a *Δhst3 Δhst4* strain the distinction between chromatin at the front and the back is blurred by the fact that all histones H3 are hyper-acetylated [69]. Under these circumstances, Ctf18 may be slowing down fork progress, leading to massive fork arrests. This phenotype would be much aggravated and become lethal in the absence of Dun1, which restrains the repression of late origin firing by Rad53. Indeed, it has been shown that fork progression is mediated by the physical association of the CMG helicase with Polε through unknown mechanisms [70] that Ctf18 could inhibit. Again, deletion Ctf18 would lead to reduced replication load and elongated elongation tracts, thus preventing massive lethality.

In conclusion, we have presented evidence for the fact that cells suffering from H3K56 hyper-acetylation are dependent on mechanisms that ensure proper regulation of their DNA replication program. We uncovered a novel, dNTPs-independent role for the protein kinase Dun1 in regulating late origin firing, a function that is also sensitive to the activity of the alternative clamp loader Ctf18. Our results provide further understanding of the effects of chromatin on origin firing and in replication at large, processes that are de-regulated in cancer cells.

## Materials and methods

### Strain construction

All strains were created in the background of YPH499/YPH500 (*MATa/MAT@ lys2-801 ade2-101 ura3-52 leu2del1 trp1del63 his3del200*) [71] and IC703/IC705 (*MATa/MAT@ his3del200 leu2del1 lys2del202 trp1del63 ura3-52 hst3del3:HIS3 hst4del2::TRP1 +pRS416-HST3*) [5]. Standard yeast molecular genetics techniques were used to create mutants. Double and triple mutations were generated by crosses and tetrad dissection in order to segregate out any spontaneous mutations. All *Δhst3 Δhst4* strains carried the complementing *pRS416-HST3* plasmid and were kept on SC plates lacking uracil to maintain the plasmid as long as necessary. Plasmid-less strains were obtained by plating on 5-FOA medium and used immediately. All strains carrying point mutations were sequenced and all strains carrying a tag were checked for the presence of a working protein.

All strains and their full genotypes can be found in S2 Table.

### Drop assay

Strains were grown in liquid SC media lacking uracil overnight, diluted 1:5 and 5μl of each dilution was spotted onto the indicated plates. Plates were incubated at the indicated temperature for 3–5 days, depending on the growth rate of the mutant in question.

### Protein extraction, Western blotting and immunoprecipitation

Cells were grown to $OD_{600}$ of 0.8–1.0, collected, washed and resuspended in 20% TCA, and protein was extracted using bead beating.

For the running of the protein gels, Acrylamide gel was prepared in varying concentrations. Running and transfer were performed in TG-SDS and TG-methanol buffers, respectively. Membranes was blocked with 1% skim milk and exposed to the suitable primary antibody (from Santa Cruz: c-Myc (9E10) and HA (F-7), Flag from Sigma Alderich (F1804)) and secondary antibody (Abcam's Goat anti Mouse, 97040) and later to a mixture of stable peroxidase and luminase (SuperSignal West Pico 34579 from Thermo) to generate final pictures.

For imaging of Dun1's phosphorylation gels were prepared as described, but were supplemented with Phos-Tag (NARD Institute, AAL-107) and 0.1mM of $MnCl_2$. Following running, gels were washed for 10 minutes with 50μl of running buffer containing 1X transfer buffer with 1mM of EDTA, followed by another 10 minute wash with only 1X transfer buffer. Gels were then put to a transfer as was previously described, but with methanol activated PVDF membrane.

## Checkpoint activation

Rad53 was tagged with HA in the *Δhst3 Δhst4 +pRS416-HST3* strain, and the relevant combination of deletions was introduced into it. Strains were then grown overnight, diluted in the morning and upon reaching mid-log phase (as was determined under a microscope, >85% of cells with small buds) either treated with 0.1% MMS or left untreated for an hour. Cells were then harvested for TCA protein extraction as described above.

## qPCR

Cells carrying a genomic degron version of *DUN1* and carrying, or not, *pRS416-HST3*, were grown to mid-log phase and arrested with 40ng/μl of alpha factor for 1.5 hours. Following that, half of the samples were incubated with 300μM of auxin for 1.5 hours. Cells were washed once with fresh media and released from arrest with 0.1mg/ml of pronase into SD medium containing HU. Samples were taken at the start of the experiment and after 80 minutes. At least two biological repeats of each experiment were carried out, each in at least three technical repeats. Cells were washed, resuspended in breaking buffer (50mM HEPES-KOH pH7.5. 140mM NaCl, 1mM EDTA, 1% triton X100, 0.1% NA-Deoxycholic acid) and glass beads and were shaken in a bead beater for 10 minutes. DNA was broken using sonication, centrifuged and the supernatant diluted with elution buffer (50mM Tris/HCl pH8, 10mM EDTA, 1% SDS) and incubated overnight at 65°C. DNA was then cleaned by centrifugation with phenol chloroform, and again with chloroform, before resuspended in ammonium acetate and 100% ethanol. DNA was then incubated at -20°C for an hour, washed with ethanol and eventually resuspended in DDW. For qPCR, DNA was diluted to 1.6ng/μl. qPCR was performed using Thermo Fast SYBR green master mix (4385614) on Applied Biosystems StepOnePlus Real Time PCR cycler.

The following primers were used:
ARS305 F agccttctttggagctcaagtg
ARS305 R tttgaggaatttctttttgaagagttg
14–705 F cggtatccacttcatctgctgcc
14–705 R tggcaacgtcgatgacgaagg
NegV F taattggctgagcgttgcatgtt
NegV R gcctctacagtaccgtggggaga

## Supporting information

**S1 Fig. Rates of spontaneous plasmid loss. (A)** Strains containing *Δade2 Δade3* and carrying plasmid *pRS316-ADE3-HST3* were grown overnight in selective media, then diluted and plated

onto YPD. Following 3 days of growth, YPD plates were replicated onto SD-Ura plates and the colonies were compared for growth between the plates. Most of the colonies that failed to grow on SD–Ura plates were also white, with a negligible percentage of red colonies. **(B)** Tetrad analysis showing that a triple deletion of *HST3*, *HST4* and *DUN1* is inviable. *An Δhst3 Δhst4 Δdun1* strain carrying a p*URA3-HST3* plasmid was crossed to an *Δhst4* strain. Diploids were streaked onto 5-FOA to lose plasmid, and sporulated. Tetrads were dissected under a micro-manipulator and their genotype was confirmed by markers.
(TIF)

**S1 Table. Genes screened in this study and the results of the screen.**
(XLSX)

**S2 Table. A list of strains and plasmids used in this study.**
(XLSX)

## Acknowledgments

The authors would like to thank the labs of Jeff Boeke, Philippe Pasero, Armelle Langronne, Karim Labib, Naama Barkai, Li Qing, Richard Kolodner, Francesc Posas and John Diffley for kindly sharing their strains and protocols with us. We thank Adi Tessler and all members of the Kupiec lab for support and ideas.

## Author Contributions

**Conceptualization:** Lihi Gershon, Martin Kupiec.

**Formal analysis:** Lihi Gershon, Martin Kupiec.

**Funding acquisition:** Martin Kupiec.

**Investigation:** Lihi Gershon, Martin Kupiec.

**Supervision:** Martin Kupiec.

**Writing – original draft:** Lihi Gershon, Martin Kupiec.

**Writing – review & editing:** Lihi Gershon, Martin Kupiec.

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
