## [Decision Letter · Decision Letter 0]

4 Nov 2020

Dear Martin,

Thank you very much for submitting your Research Article entitled 'Novel roles for Dun1 and Mrc1 in the regulation of origin firing upon hyper-acetylation of H3K56' to PLOS Genetics. Your manuscript was fully evaluated at the editorial level and by independent peer reviewers. The reviewers appreciated the attention to an important problem and the thorough genetic analyses done, but requested additional experiments to further support the conclusions as well as textual modifications.  A summary figure presenting the inferred models might also be helpful.  Based on the reviews, we will not be able to accept this version of the manuscript, but we would be willing to review again a much-revised version. We cannot, of course, promise publication at that time.

If you decide to revise the manuscript for further consideration at PLOS Genetics, please aim to resubmit within the next 60 days, unless it will take extra time to address the concerns of the reviewers, in which case we would appreciate an expected resubmission date by email to plosgenetics@plos.org.

[LINK]

Please do not hesitate to contact us if you have any concerns or questions.

Yours sincerely,

Sue Jinks-Robertson, Ph.D.

Associate Editor

PLOS Genetics

Gregory P. Copenhaver

Editor-in-Chief

PLOS Genetics

Reviewer's Responses to Questions

**Comments to the Authors:**

Reviewer #1: This is beautiful piece of work of the Kupiec lab showing using classical yeast genetics that Dun1 has a so far unknown function controlling Rad53 phosphorylation and in turn late origin firing, as a way to explain the lethality caused by dun1 in hst3 hst4 genetic background that is suppressed by rad53. A similar study is made for mrc1 mutants, but in this case the molecular explanation for the lethality caused by triple mutations seems different and not caused by suppression of late firing. The latter, provides a good example of the complexity of events that can cause lethality of hst3 hst4 mutants and the need to understand the molecular events generated in these double mutants that can compromise viability. It is a manuscript appropriate for PLoS Genetics that researchers in the field will find highly inspiring. I just make a number of suggestions and request few controls that will make the conclusions more convincing by going a step ahead and showing by a minimum molecular analysis the validity of the model proposed.

Title. Is it really appropriate? I am not sure the manuscript shows that Mrc1regulates origin firing.

Introduction. It is too long. It could be considerable reduced, in particular page 4 which is more adequate for a high school reader.

Some parts of the Results section can be shortened. Authors write the manuscript reasoning nicely every experiment. However, in a number of cases the premises on which the experiments were based did not fit with the results obtained. This is the case showing that the viability does not depends on the levels of PCNA in chromatin (page 10). There is no need for so many explanations for a negative result.

Similar suggestion on the negative role of Dun1 as a regulator of RNRs’ negative regulators in page 11.

Similar suggestions about the negative result showing no involvement of chromatin remodelers in viability, this part being unnecessarily long (pages 11 & 12)

Westerns in Figs. 1C and 4C do not show the WT. It needs to be shown.

Checkpoint activation of Rad53 after H3K56 hyper-acetylation. ∆rad53 suppresses dun1 lethality but not mrc1. Perhaps ∆mrc1 leads to a lethal intermediate, since Mrc1, in contrast to Dun1 that is a kinase and a regulator, Mrc1 is a structural protein working at forks the absence of which can alter the output of possible RF collapses causing toxic intermediates that would lead to lethality. Do authors consider this to be an option? Discuss.

Hst3 hst4 mutants have been shown to have other phenotypes such as high levels of Rad52 foci and high recombination (Alvaro et al PLoS Genet 2007), synthetic lethality with rad52 (Celic et al, Genetics 2008) and defective sister chromatid recombination repair (Munoz et al, PLoS Genet 2013). It seems that, as likely expected for mutants affecting chromatin, hst3 hst4 creates a pleiotropic vulnerability that makes cells be dependent on functions non-essential in wt cells, like dun1, rad52, mrc1, and more, as summarized in Supp TAble 1. Authors could discuss these other options, in particular for the mrc1 mutation. Wouldn’t make sense to think that lethality caused by Mrc1, which, in contrast to Dun1, plays a physical role at forks and likely at collapsed fork during repair, is related to the accumulation of unrepaired collapsed forks or DSB intermediates? It would help the article to expand Discussion taking into account these facts.

Once authors infer that it is the ability of constitutively fire late origins in ∆rad53 cells what rescues the lethality caused by ∆Dun1, wouldn’t it be necessary to check this? Authors could check that one late origin controlled by Rad53 fires replication at early S phase in dun1 cells via ChIP with a pulse of BrdU using appropriate yeast strains able to incorporate BrdU. Authors should try to add direct evidence to their model on the regulation of origin firing, if they want to be that explicit in title and conclusions.

While the use of FOA to check the capability of cells losing the DUN1-containing plasmid is perfectly valid, authors could show the tetrad analysis of hst3 hst4 x dun1 and hst3,4 x mrc1, to see that triple mutant spores do not grow. Since mutants of chromatin states usually have pleiotropic phenotypes, affecting also transcription whether or not by direct means, it would be nice to see the phenotype of the hst mutations in FOA-free media.

It would important to show whether H3K56 mutations mimicking acetylated H3K56 (Q), is dead with dun1, but not mutations mimicking non-acetylated (R or A), to support the conclusions on the specificity of H3K56 acetylation.

Reviewer #2: In this work, the authors describe synthetic lethal or sick (SS) interactions between either dun1 or mrc1 mutants and hst3∆ hst4∆ mutant. In addition, they uncovered several mutants suppressing hst3∆ hst4∆ mutant growth defect at higher temperatures. Besides known hst3∆ hst4∆ suppressors, such as asf1, they uncovered ctf18∆, rad53∆ and dbf4-3A sld3-38A. These three suppressors can exert their suppressive effects also in cells containing hst3∆ hst4∆ dun1∆. Only ctf18∆ and dbf4-3A sld3-38A has suppressive effect in cells containing hst3∆ hst4∆ mrc1∆.

To understand the SS (dun1∆ or mrc1∆) and suppressive effects (ctf18∆, rad53∆ and dbf4-3A sld3-38A) on hst3∆ hst4∆, the authors performed additional genetic analyses, which ruled out the involvement of cohesion, alteration of PCNA level on chromatin, histone, and dNTP levels.

The authors further used separation of function alleles of dun1 and mrc1, and ctf18 to suggest that Dun1 interaction with Rad53 is not important for the SS with hst3∆ hst4∆, while loss of Ctf18 or Mrc1 binding to Pol epsilon can be relevant.

One strength of the work is the findings of additional hst3∆4∆ genetic interactions. Another strength is the finding that the negative interactions between Dun1∆ and hst3∆4∆ is unrelated to dNTP regulation. The authors suggest that this can mean that Dun1 has a role in origin firing independent of Rad53. The weakness is the lack of mechanistic insights into any of these interactions - genetic interactions can have more than one interpretation, thus leaving open many possibilities. Following up on one of the many speculations described in the work at a mechanistic level can strengthen the paper. For example, there are many tools available to test whether Dun1 can counter Rad53 in origin firing regulation or if Ctf18 indeed rescues the DNA replication defects seen in hst3∆4∆, hst3∆4∆mrc1∆, or hst3∆4∆dun1∆ cells. Also, most genetic tests were done using a plasmid carrying Hst3. For the most critical conclusions, one shall use integrated alleles and degron alleles to verify the findings, as that some of these genetic interactions could be influenced by plasmid instability.

Reviewer #3: This manuscript reports on the genetic relationships between histone acetylation, DNA replication and various aspects of checkpoint signalling in budding yeast. The authors use a robust genetic assay to examine synthetic lethal as well as suppressive relationships in a background of hyperacetylation of new histones. The topic is an interesting one, as the acetylation and subsequent deacetylation of H3 are both required for genome stability, but the reason behind this is not well understood. In this study, the authors describe an essential function of the two checkpoint factors Dun1 and Mrc1 in situations where H3 deacetylation is impeded. They find that the essential function can be suppressed by deletion of another component of the replication fork, Ctf18, and they exclude a series of possible scenarios as reasons, e.g. interactions with the replicative polymerase epsilon or PCNA loading/unloading. Having excluded these factors, they go on an examine a set of checkpoint factors, where they find complicated relations that on one hand differentiate between Dun1 and Mrc1 (i.e. showing non-identical phenotypes) and on the other hand identify a contribution of the checkpoint effector kinase Rad53 as well as an important function of regulating origin firing.

Overall, the experiments shown here appear clean, and the assay they use gives robust data. I have no objections to this experimental approach, as it provides reliable results. I also appreciate the systematic manner in which the authors exclude some potential factors that could be invoked as a model. However, at the end of the day, I am left wondering how to integrate the positive data into a model that makes sense. As it stands, the authors have identified a series of potentially interesting genetic relationships that indicate a role of Dun1 beyond its known functions, but neither the relevant target nor the physiological consequences are clear from the data presented. The notion that Dun1 and Mrc1 share many, but not all phenotypes, also doesn't help to gain any mechanistic insight. As such, the manuscript unfortunately remains rather descriptive. This also becomes apparent in the abstract and the discussion, where the authors pretty much summarize their results, but do not really provide an interpretation.

Some specific points:

1. I did not really understand the section about the various checkpoint factors (starting line 311). They state that according to the model, all the checkpoint factors should be essential in the hst3/4 mutant, but further down that page, they argue about suppressive effects by rad9 and/or mrc1. This needs to be better explained.

2. Lines 383-386: I am not sure how the authors can conclude that Dun1 directly regulates origin firing from the notion that derepression of origin firing rescues viability.

3. There are multiple instances of sloppy expressions, genetically speaking. E.g.:

- line 241: "nor do they restore viability to Δdun1" - what is "they": the proteins or their mutants?

- line 275: "It was also found that they weaken the association with Pol2 subunit of Polε" - again, what is "they": the aa positions or mutants thereof?

- line 301: "the synthetic lethality of Δdun1 and Δmrc1" - this would implicate a synthetic lethality between dun1 and mrc1, but I think the authors mean synthetic lethality between either dun1 and hst3/4 or mrc1 and hst3/4.

Minor point:

4. Figure 3A is poorly drawn and quite confusing.

5. "dNTPs levels" should probably be "dNTP levels" or "levels of dNTPs"

**Have all data underlying the figures and results presented in the manuscript been provided?**

Reviewer #1: Yes

Reviewer #2: Yes

Reviewer #3: Yes

PLOS authors have the option to publish the peer review history of their article (what does this mean?). If published, this will include your full peer review and any attached files.

Reviewer #1: No

Reviewer #2: No

Reviewer #3: No

---

## [Decision Letter · Decision Letter 1]

21 Jan 2021

Dear Martin:

Thank you very much for submitting of revision of your Research Article entitled 'A novel role for Dun1 in the regulation of origin firing upon hyper-acetylation of H3K56' to PLOS Genetics.  The reviewers were enthusiastic about the more focused version, and had only a few additional comments.  Most were minor, with the exception of whether the qPCR done measure late origin replication is really a measure a origin firing.

We therefore ask you to modify the manuscript according to the review recommendations. Your revisions should address the specific points made by each reviewer.

[LINK]

Best regards,

Sue

Sue Jinks-Robertson, Ph.D.

Associate Editor

PLOS Genetics

Gregory P. Copenhaver

Editor-in-Chief

PLOS Genetics

Reviewer's Responses to Questions

**Comments to the Authors:**

Reviewer #1: Authors responded to my comments in a satisfactory manner. The manuscript is much improved. and appropriate for PLoS Genetics. However when introducing in Discussion possible new interpretations, as suggested, authors should cite the three references I indicated in my review. Although one is already cited in anorther context, it is relevant to mention the references to support the alternative explanation and because they are relevant for this study.

Reviewer #2: The authors included new qPCR data at two ARS sites to address origin firing in cells containing a Dun1 degron allele. Unfortunately, this assay can be influenced by factors other than origin firing such as passive replication. More appropriate assays would be 2D gel analyses or ChIP-seq. In the case that these assays are difficult to perform given the pandemic situation, one could at least show FACS profiles, which give a low-resolution readout of genome replication, and qPCR (DNA copy number) data across multiple early vs. late ARSs (not just two), as well as include other possible interpretation of the dun1 effects. I could not see any new data without using the Hst plasmid. please provide.

Reviewer #3: By focussing more on Dun1 rather than comparing Dun1 with Mrc1 and by re-writing relevant sections of the discussion, the manuscript has now become much clearer. I am now happy with the revised version. It might be worth carefully reading over the text once more for a few remaining grammatical issues.

**Have all data underlying the figures and results presented in the manuscript been provided?**

Reviewer #1: Yes

Reviewer #2: Yes

Reviewer #3: Yes

PLOS authors have the option to publish the peer review history of their article (what does this mean?). If published, this will include your full peer review and any attached files.

Reviewer #1: No

Reviewer #2: No

Reviewer #3: No

---

## [Editor Report · Decision Letter 2]

1 Feb 2021

Dear Martin,

We are pleased to inform you that your manuscript entitled "A novel role for Dun1 in the regulation of origin firing upon hyper-acetylation of H3K56" has been editorially accepted for publication in PLOS Genetics. Congratulations!

Yours sincerely,

Sue

Sue Jinks-Robertson, Ph.D.

Associate Editor

PLOS Genetics

Gregory P. Copenhaver

Editor-in-Chief

PLOS Genetics

**Data Deposition**

http://datadryad.org/submit?journalID=pgenetics&manu=PGENETICS-D-20-01560R2

**Press Queries**

---

## [Editor Report · Acceptance letter]

13 Feb 2021

PGENETICS-D-20-01560R2 

A novel role for Dun1 in the regulation of origin firing upon hyper-acetylation of H3K56 

Dear Dr Kupiec, 

We are pleased to inform you that your manuscript entitled "A novel role for Dun1 in the regulation of origin firing upon hyper-acetylation of H3K56" has been formally accepted for publication in PLOS Genetics! Your manuscript is now with our production department and you will be notified of the publication date in due course.

With kind regards,

Alice Ellingham

PLOS Genetics

On behalf of:
